# Child and adolescent food insecurity in South Africa: A household-level analysis of hunger

**Siluleko Mkhize**[1], **Elena Libhaber**[2], **Ronel Sewpaul**[3], **Priscilla Reddy**[3,4], **Laurel Baldwin-Ragaven**[1]*

**1** Department of Family Medicine and Primary Care, School of Clinical Medicine, Faculty of Health Sciences, University of the Witwatersrand, Johannesburg, South Africa, **2** School of Clinical Medicine and Health Sciences Research Office, Faculty of Health Sciences, University of the Witwatersrand, Johannesburg, South Africa, **3** Health and Wellbeing, Human and Social Capabilities Division, Human Sciences Research Council, Cape Town, South Africa, **4** Faculty of Health Sciences, Nelson Mandela University, Port Elizabeth, South Africa

* laurel.baldwin-ragaven@wits.ac.za

**Data Availability Statement:** The final de-identified anonymised data set used in this study is provided as a "Supporting information" file in this manuscript (S1 Database). A data sharing agreement was signed with the Human Sciences

## Abstract

Food insecurity impacts childhood nutritional status, physical and cognitive development, and increases lifetime risk for chronic disease. Previous South African studies have examined hunger at the sub-national level without a specific focus on children and adolescents. This study determines the national prevalence of childhood food insecurity, from birth to adolescence, and identifies factors associated with hunger within the household. Individual and household-level data were extracted from the South African National Health and Nutrition Examination Survey (SANHANES-1). Prevalence of food insecurity was assessed using the Community Childhood Hunger Identification Project (CCHIP) index. Multinomial logistic regression analyses were conducted on all households (with and without children) to determine the predictors of food insecurity, with additional analyses adjusting for child dependency and sociodemographic characteristics of household heads in households with children. Of 5 098 households surveyed, 68.6% had children and adolescents present (0– 19 years). Of these households, 32.5% (95% Confidence Interval [CI]: 29.5–35.7) were experiencing hunger and 26.3% (95% CI: 23.9–28.8) were at risk of hunger. Among all the households, significant associations for experiencing hunger were the presence of children and adolescents: Adjusted Odds Ratio (AOR) = 1.68 (95% CI: 1.12–2.53); being female-headed: AOR = 1.53 (95% CI: 1.21–1.94) and informally-located; AOR = 1.6 (95% CI: 1.07– 2.43). Of the racial groups, having a non-African household head, Coloured: AOR = 0.29 (95% CI: 0.19–0.44) and White/Indian/Asian: AOR = 0.12 (95% CI: 0.04–0.33) conferred lower odds of experiencing hunger; and, the household head having secondary/ tertiary education conferred lower odds of experiencing hunger; AOR = 0.40 (95% CI: 0.28– 0.56) as well as being at risk of hunger; AOR = 0.69 (95% CI: 0.52–0.92). Receiving social grants, pensions, or remittances more than doubled the odds of experiencing hunger; AOR = 2.15 (95% CI: 1.49–3.09). After adjusting for child dependency in households with children, having at least one older child (age 15–19 years old) did not change the odds of food insecurity. In summary, only 41% of South African households with children and adolescents were food secure. The associations between household head sociodemographics,

Research Council (HSRC), the curator of the data collected in the 2012 ANHANES survey. Adult SANHANES data are available through registered access from the HSRC data repository at: http://curation.hsrc.ac.za/Datasets-XKAHAA.phtml. Child data are under third party rights and can be accessed upon reasonable request by contacting datahelp@hsrc.ac.za.

**Funding:** SM received funding from the South African National Research Foundation (NRF): Freestanding, Innovation and Scarce Skills Development Fund Masters and Doctoral Scholarships (Reference: SFH170525233408 and MND190719457401) to support his postgraduate studies enabling him to pursue this research. SM received additional funding from the Carnegie Corporation of New York, in the form of a grant (G-17-55194) to complete his degree.

**Competing interests:** The authors declare that no competing interests exist.

household location and size on household food insecurity indicate a need for multi-sectoral interventions to bolster sustainable food systems for households with children and adolescents and to improve public protections for female-headed, African-headed and informally-located households dependent on social grants.

## Introduction

Food insecurity has emerged as a major public health concern; and, the obligation to realise the "*right to food and basic nutrition*" is enshrined in both the United Nations Charter [1] and the South African Constitution [2]. The Food and Agriculture Organization (FAO) of the United Nations states that "food security exists when all people, at all times, have physical, social and economic access to sufficient, safe and nutritious food that meets their dietary needs and food preferences for an active and healthy life" [3], emphasising that the availability of food must be beyond mere subsistence [4]. In Sub-Saharan Africa, one in four people experienced undernutrition in the year 2017. This represents about one-third of the global population estimated to have suffered from chronic hunger in the same year [5]. According to actuarial projections, Africa will be home to 90% of the world's poorest by 2030 [6].

In 2017, 6.8 million South Africans were affected by hunger, representing a 50% decline since 2002; however, at the household-level, the burden of food insecurity remains largely unabated [7]. Moreover, a considerable but undocumented proportion of these households comprise children and adolescents who are vulnerable to food deprivation and hunger [4]. Indeed, it has been suggested that the presence of children in households may confer additional "*pressure of mouths to feed*" [7, 8].

Throughout the world, household food insecurity is associated with malnutrition, especially stunting and wasting among children under five years old [9]. Nationally, Lake et al. [10] reported that approximately half of South African children under age five were malnourished; and, severe-acute and moderate-acute malnutrition accounts for between 4% to 25% of child mortality [10, 11]. Such implications of food insecurity on health and nutrition have been observed not only in children under five, but among older children as well. The largest national census of South African households conducted in 2011/12, the South African National Health and Nutrition Examination Survey (SANHANES), reported that the highest prevalence of wasting among children was actually among boys and girls aged 10–14 years (5.6% and 2.5% respectively); and, underweight was most prevalent among boys and girls in the 7–9 year-old age group (8.6% and 4.0% respectively) [12]. In older children, the South African Demographic and Health Survey (SADHS) showed that 27% and 9% of adolescent girls and boys aged 15–19 years respectively, were overweight or obese [13], highlighting the double burden of malnutrition across childhood and adolescence in South Africa. Importantly, hunger has profound health consequences that not only affect physical growth and cognitive development in childhood [14, 15], but endure throughout the life-course into adulthood [16, 17] and extend into subsequent generations [18, 19]. Two longitudinal cohort studies of the consequences of hunger that began in the throes of the Second World War, the Dutch Winter Famine [19] and the siege of Leningrad [18, 20] indicate that the effects of intra-uterine exposure to food insecurity can traverse through multiple generations, predisposing to a "thrifty phenotype" among progeny. Bjerregaard, et al. [21] found that another form of malnutrition, obesity in childhood (measured at ages 7 and 13 years), significantly increased the hazards of developing type-2 diabetes in late adulthood among Danish men. Conversely, reversal of obesity in adolescence

reduced the risk by 4%, a finding that may be explained, in part, by food insecurity given that obesity and type-2 diabetes are primarily diet-sensitive adverse health outcomes [15].

In South Africa, the Community Childhood Hunger Identification Project (CCHIP) index was used to estimate the prevalence of food insecurity nationally between 1999 and 2008 in three different studies: the National Food Consumption Surveys of 1999 and 2005, and the South African Social Attitudes Survey of 2008 [22]. A review of these surveys, which included children from 1–9 years old with at least one adult in the household, indicated a marked decline in household food insecurity from 52.3% in 1999 to 25.9% in 2008 [22]. These studies looked at a segment of children (those 1–9 years old), thus perhaps underestimating the true extent of child and adolescent food insecurity [23, 24]. Much of what is known about the sociodemographic correlates of food insecurity in South Africa has been gleaned from sub-national cross-sectional studies. Food insecurity measured using the Household Food Insecurity Access Scale (HFIAS) indicated a gradient across the rural-urban continuum of South African towns, with households in rural areas reporting higher scores, and therefore more food insecurity, than peri-urban households (households located in urban-informal settlements) [25]. Households that are reliant on private food charities and government-assisted social grants are at an increased risk of food insecurity [26]. It is speculated that while social grants are regular and predictable, they are, however, not sensitive to the changing needs of a household [26, 27]. The gender of the house-hold head is also an important factor predicting household vulnerabilities [9], with the highest prevalence of food insecurity in South Africa reported among female-headed households [12]. In an analysis of international data, women are more likely to experience food insecurity in comparison to men, which is attributed mainly to disparities in income, education and access to social networks [28]. Even with increased employment rates among women, the income differentials between women and men has meant that poverty remains strongly gendered. Research into possible reasons for the gendered nature of food insecurity in South Africa include decades of migrant labour practices disrupting family structures, and household size [7]. Female-headed households are larger, on average, than male-headed households and include more dependent children and adolescents, as well as are less likely to have any household member working [11].

There is a paucity of nationally representative data to assess the granular detail about food insecurity across the country. The critical need to address hunger in childhood necessitates particular attention to factors affecting food insecurity in households with children and adolescents—such as the household size, household head sociodemographics and dependency status of children—so that nutrition-sensitive interventions can be developed and/or strengthened [29]. Such interventions would address the underlying determinants of child hunger and include, for example, multi-sectoral approaches to social safety netting through minimum guaranteed income and/or cash transfers; financial, educational and parenting support to female-headed households with children; and bolstering health services and sustainable agricultural practices [5, 29]. Such multi-pronged approaches are necessary to mediate the short-term, long-term and inter-generational adverse health consequences of food insecurity on childhood health. Therefore, drawing from a large-scale population-based sample of households, the SANHANES-1, we set out to determine the prevalence of food insecurity in households with and without children as well as to assess associations between the sociodemographic characteristics of the household head and degrees of hunger.

## Methods

### Study design

In 2019–2020, we conducted a secondary data analysis of the SANHANES-1 (2011/12) [12]. The SANHANES-1 is a cross-sectional nationally representative sample of households that

employed a multi-stage disproportionate, stratified cluster sampling approach based on the 2001 census Enumeration Areas (EAs). The first 20 households were sampled from each of the 500 EAs yielding a sample of 10 000 households. Ensuring national representativeness, the SANHANES-1 EA sampling was layered by type of locality (urban/rural; formal/informal), and province (there are nine in South Africa) as well as by "race" in formal urban areas. Using race as a variable allows investigation of ongoing health disparities that have endured post-apartheid. In this way, the SANHANES captures the sociodemographic and economic profile of the country. Additional sample selection details have been published elsewhere [12]. In our study, we used the "household" as the unit of analysis. Statistics South Africa (Stats-SA) defines a household as "a group of people who live together, eat together and share resources, or a single person who lives alone" [30]. In the SANHANES-1, household membership was attributed to persons who occupied the same dwelling and slept in the household for at least four nights a week. The household head was identified as a physically present member who was designated internally as the head of that household, and also served as the main respondent for household-level data acquisition.

## Data extraction and study population

The SANHANES-1 database includes people of all ages residing in households in South Africa. For the present study, only occupied households were included (S1 Fig). Five separate but inter-related SANHANES-1 questionnaires were administered at the household-level by trained field workers capturing information on everyone living in that household (between 1 and 20+ people). The variables of interest for this study were extracted from each of the five questionnaires: visiting point (household) questionnaire (administered to the household head); child questionnaire (0–14 years old); child clinical examination form (0–14 years old); adult questionnaire (administered to anyone ≥ 15 years old); and the adult clinical examination form (≥ 15 years old). After extraction, these data were entered into four separate newly created databases. Children, whom we defined as aged 0–19 years old in order to include the adolescent population in our sample, and adults aged ≥20 years old were extracted from both the child (<15 years) and adult (≥15 years) individual-level datasets, respectively. This age selection aligns with global and national initiatives that target child and adolescent public health and nutrition [31–35]. After merging data for individuals aged 15–19 years with data from those 0–14 years old, thereby reclassifying everyone aged 0–19 years old as "children", we then reconciled them back into their corresponding households. The final sample that we analysed included 5 098 households and their occupants, both adults and children/adolescents. S1 Fig illustrates the database management, organisation and integration of the different household and individual-level variables from the five SANHANES-1 databases.

## Variables

**Household composition and sociodemographic parameters.** At the household-level, the parameters included were household size, the number of adults, the number of children occupying a household and child dependency status. Two variables were created to reflect household dependency status: those with children aged 0–14 years only (yes or no) and those with at least one child 15–19 years old (yes or no), who could theoretically contribute to overall household income given the legal age at which one can start working in South Africa [36]. Age, self-reported sex (male and female), self-reported race (African, Coloured, White/Indian/Asian), educational attainment (no schooling, primary, secondary, tertiary/higher degrees), marital status (married, living together/civil union, never married, widowed, separated/divorced) and the main source of income (salaries/wages, social grants/remittances/pensions, sale of products

and services, no income) of all household heads were also included. Locality (urban, rural, formal and informal) was pre-assigned to the household from the EA sampling.

**Household food insecurity.** Food insecurity was measured using the CCHIP index, a validated tool for the assessment of food insecurity at the household-level [37]. While five of the eight questions are child-referenced, making the CCHIP a specific tool for measuring *childhood* food insecurity, the CCHIP was also administered to households without children, generalising the child-referenced questions to everyone in the household. A score of ≥5 affirmative responses indicate the presence of hunger, understood as "*experiencing hunger*" and recorded as such. A score of 1–4 affirmative responses indicates that the household is "*at risk of hunger*". Lastly, a score of zero indicates that the household is "*food secure*".

## Statistical analyses

STATA software, version 16.1 (STATA Institute Inc., College Station, TX, USA) was used for database management and statistical analyses. Results are presented as frequencies and percentages for categorical variables and Median [IQR] or arithmetic Mean ± SD in the case of continuous variables. Comparisons between households with and without children were performed using Mann-Whitney U tests for continuous non-normally distributed data. Categorical variables were compared using Chi-Square tests. All analyses were weighted to offset for the over-allocation of EAs in areas where Indian, Coloured or White racial groups prevailed, to ensure that the minimum required sample size in those minority groups were obtained. All analyses were performed using *svy* function in STATA, incorporating sample weights and stratified cluster sampling design to provide estimates. Weighted prevalence of food security in households with children, stratified by sociodemographic characteristics of the household heads was ascertained from frequencies and percentages; however, age differences in prevalence estimates were computed from a Kruskal-Wallis test with a Bonferroni *post hoc* correction to adjust for multiple comparisons. To improve statistical power in the regression models, locality was dichotomised to formal vs informal; educational attainment (primary, secondary/tertiary/higher degrees and no schooling); marital status (married, never married, living together/civil unions, and widowed/separated/divorced). Weighted univariable and multivariable multinomial logistic regression analyses with robust standard errors were carried out to determine factors associated with being "at risk of hunger" and "experiencing hunger", where the "food secure" category was used as the referent group for comparison in all households, and separately in households with and without children. In households with children, we conducted additional univariable multinomial logistic regressions to examine associations between food security and child dependency, and between food security and female-headed households. A multivariable multinomial logistic regression was carried out to adjust for all of the sociodemographic factors related to the household. Statistical significance was set at p<0.05. Multivariable multinomial logistic regression for "at risk of hunger" and "experiencing hunger" using adjusted odds ratios with their 95% confidence intervals (CI) as measures of association are presented as forest plots, which were produced using GraphPad Prism, version 7.00 for Windows (GraphPad Software, La Jolla, California USA).

## Ethical considerations

The original SANHANES-1 (2011/12) received ethics approval from the Research Ethics Committee (REC) of the Human Sciences Research Council (HSRC) (REC 6/16/11/11). The present secondary analysis received ethics approval from the Human Research Ethics Committee (HREC) (Medical) of the University of the Witwatersrand, Johannesburg, South Africa (Clearance certificate number: M180775).

## Results

### Household composition, living arrangements and main source of income of household heads

Table 1 shows the characteristics of South African households with and without children, and the living arrangements of the heads of households. Of the 5 098 households with complete CCHIP scores, 3 499 (68.6%) households contained children aged 0–19 years. Households

**Table 1. Characteristics of occupied South African households with and without children, and sociodemographic characteristics of household heads.**

| | All Households N = 5 098 | | p value |
|---|---|---|---|
| | With children n = 3 499 | Without children n = 1 599 | |
| **Household characteristics** | | | |
| Household size (n = 21 932) | 5 [4–6] | 2 [1–2] | <0.001[&] |
| No. of Adults, ≥20 years (n = 12 748) | 2 [2–3] | 2 [1–2] | <0.001[&] |
| No. of Children, ≤19 years (n = 9 183) | 2 [1–3] | - | - |
| **Gender (n = 5 092)** | | | |
| Male | 1 532 (31.77) | 899 (15.76) | <0.001[#] |
| Female | 1 963 (40.44) | 698 (12.02) | |
| **Age, years (n = 5 093)** | | | |
| Mean ± SD | 49.22 ± 14.67 | 52.07 ± 15.91 | <0.001[&] |
| **Race (n = 5 090) †** | | | |
| African | 2 465 (60.88) [a] | 849 (19.86) | <0.001[#] |
| Coloured | 659 (6.86) [b] | 263 (2.48) | |
| White/Indian/Asian | 373 (4.54) [ab] | 481 (5.39) | |
| **Locality (n = 4 889) †** | | | |
| Urban formal | 1 838 (39.25) [ab] | 892 (16.15) | |
| Urban informal | 464 (7.1) [cd] | 155 (2.54) | <0.001[#] |
| Rural formal | 359 (4.16) [ace] | 222 (6.3) | |
| Rural informal | 838 (22.82) [bde] | 121 (1.68) | |
| **Marital status (n = 4 998) †** | | | |
| Married | 1 678 (34.68) [a] | 687 (11.15) | |
| Living together/civil union | 310 (6.1) [bc] | 73 (1.28) | |
| Never married | 757 (16.02) [abd] | 486 (9.5) | <0.001[#] |
| Widowed | 522 (11.38) [d] | 236 (4.07) | |
| Separated/divorced | 156 (3.9) [c] | 93 (1.93) | |
| **Source of income (n = 4 608)** | | | |
| Salaries or wages | 1 330 (29.52) | 591 (11.00) | |
| Social grants/pensions/remittances | 1 006 (24.7) | 498 (9.49) | 0.381 |
| Products and services | 138 (3.33) | 78 (1.71) | |
| No income | 644 (13.98) | 323 (6.27) | |

Data presented as median [IQR]; arithmetic mean ± SD and n (%): unweighted (n) and weighted (%).

[&]Denotes p values obtained from Mann-Whitney U tests when comparing households with and without children for continuous variables (household size, number of adults and age of household head).

[#]Denotes p values obtained from overall Chi-Square tests when comparing households with and without children for categorical variables (gender, race, locality, and marital status).

†Denotes results from multiple 2x2 comparisons with Bonferroni correction: p<0.05 if categories share the same letter (a, b, c, d, and e) when comparing households with and without children. Specifically, for race and locality, p<0.0001 for African vs White/Indian/Asian, Coloured vs White/Indian/Asian, urban formal vs rural formal, and urban formal vs rural informal. For marital status, p = 0.001 in married vs never married and living together vs never married, p = 0.036 in Separated/divorced vs living together/civil unions and lastly p = 0.013 in never married vs widowed.

with children were occupied by 9 505 adults and 9 184 children, while those without children were occupied by 3 243 adults. Households with children were 2.5 times larger than those without children (p<0.001). Forty percent of all households in South Africa contained children and were female headed. Of the households containing children, nine were child-headed. When comparing households with and without children, the proportions of African- and Coloured-headed households were not different (p = 0.417) and, no differences between urban formal and urban informal settings were found following two-by-two comparisons (p = 0.361). Furthermore, there was no statistically significant difference in the main source of income for households with and without children (p = 0.381).

### Prevalence of food insecurity in households with children

In households with children, the prevalence of *experiencing hunger* and being *at risk of hunger* was 32.5% (95% CI: 29.5–35.7) and 26.3% (95% CI: 23.9–28.8), respectively (Fig 1). Table 2 shows the weighted prevalence of food insecurity stratified by the household head sociodemographic characteristics. Households in which the household head was female, older, of African race, unmarried, not having a formal education, dependent on social grants/pensions/remittances and having no income showed a higher prevalence of experiencing hunger or being at risk of hunger (p<0.001). Further analysis of the collapsed locality categories (formal vs informal) yielded the following: 37.2% of households with children were informally located; and, of these, only 28.9% were food secure. In contrast, of the 62.9% of households with children located in formal areas, 53.6% were food secure (p<0.001).

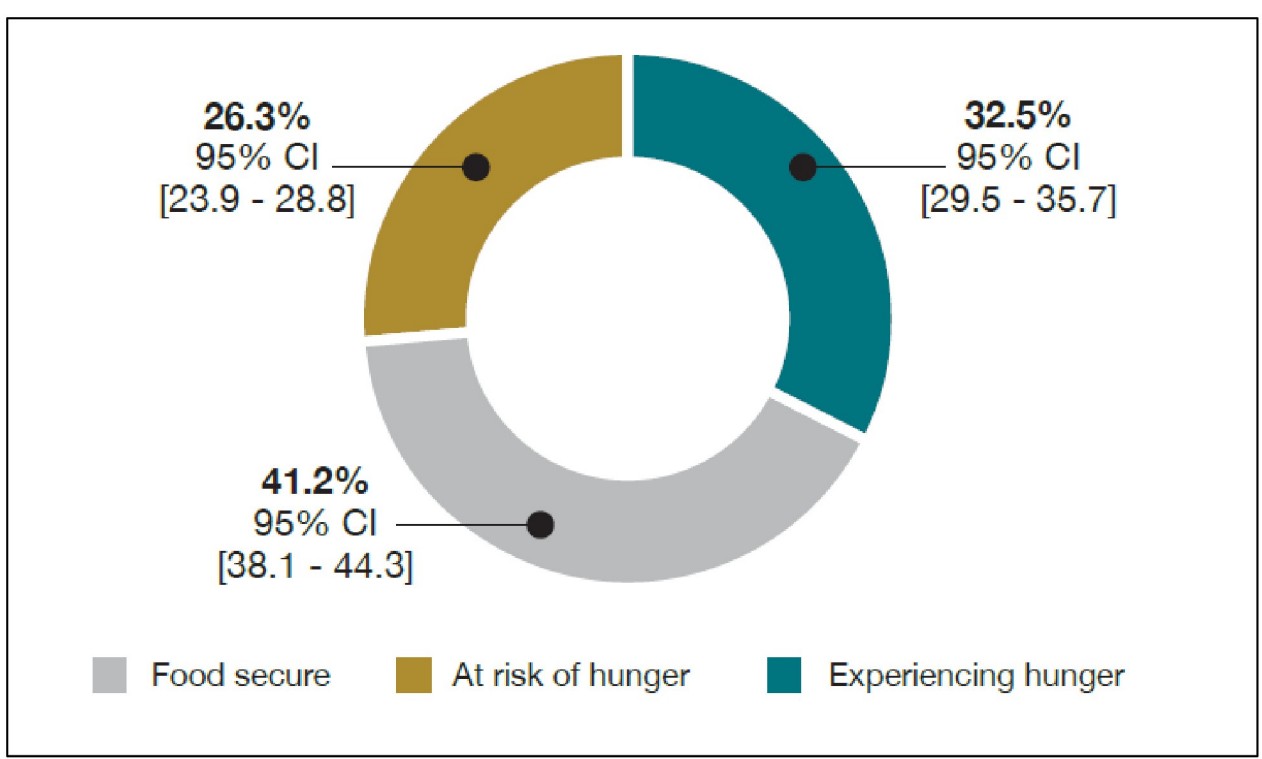

**Fig 1. National prevalence of food insecurity in households with children.**

**Table 2. Weighted prevalence of household food security, stratified by sociodemographic characteristics of designated household heads of South African households with children.**

| | Total N | Food secure n (%) | [†]At risk of hunger n (%) | [‖]Experiencing hunger n (%) | p value[#] |
|---|---|---|---|---|---|
| **Gender (n = 3 495)** | | | | | |
| Male | 1 532 | 813 (22.16) | 356 (11.09) a | 363 (10.75) a | <0.001 |
| Female | 1 963 | 739 (19.07) | 502 (15.13) a | 722 (21.8) a | |
| **Age, years (n = 3 499)** | | | | | |
| Mean ± SD | 3 499 | 47.87±13.85 [*] | 49.76±15.15 | 50.70±15.25 | 0.001 |
| **Race (n = 3 497)** | | | | | |
| African | 2 465 | 845 (30.18) | 662 (23.16) ab | 958 (30.89) ab | 0.001 |
| Coloured | 659 | 388 (5.85) | 155 (22.6) a | 116 (1.38) ac | |
| White/Indian/Asian | 373 | 319 (5.16) | 43 (0.85) b | 11 (0.27) bc | |
| **Marital status (n = 3 423)** | | | | | |
| Married | 1 678 | 899 (23.43) | 375 (12.3) a | 404 (12.39) abc | 0.001 |
| Living together/civil union | 310 | 112 (3.16) | 80 (22.9) | 118 (3.01) a | |
| Never married | 757 | 256 (6.85) | 212 (6.25) a | 289 (9.13) b | |
| Widowed | 522 | 196 (5.69) | 134 (3.98) | 192 (6.12) c | |
| Separated/divorced | 156 | 61 (1.96) | 44 (1.38) | 51 (2.07) | |
| **Educational attainment (n = 3 499)** | | | | | |
| Primary | 820 | 243 (6.48) | 223 (6.28) a | 354 (10.34) ab | <0.001 |
| Secondary | 1 524 | 762 (18.27) | 390 (12.11) b | 372 (11.49) acd | |
| Tertiary/higher degree | 353 | 280 (8.4) | 50 (1.81) abc | 23 (0.81) bce | |
| No schooling/other | 802 | 268 (8.05) | 197 (6.05) c | 337 (9.91) de | |
| **Source of income (n = 3 118)** | | | | | |
| Salaries or wages | 1 330 | 774 (22.93) | 285 (9.32) ab | 271 (9.01) ab | <0.001 |
| Social grants/pensions/remittances | 1 006 | 329 (10.69) | 266 (9.34) a | 411 (14.5) a | |
| Sale of products and services | 138 | 71 (2.11) | 37 (1.46) | 30 (1.08) c | |
| No income | 644 | 208 (5.54) | 174 (5.86) b | 262 (8.15) bc | |
| **Locality (n = 3 499)** | | | | | |
| Urban formal | 1 838 | 1 049 (27.85) | 410 (12.9) ab | 379 (12.77) abc | <0.001 |
| Urban informal | 464 | 131 (2.59) | 140 (3.11) a | 193 (3.98) a | |
| Rural formal | 359 | 128 (2.18) | 80 (1.19) | 151 (2.3) b | |
| Rural informal | 838 | 245 (8.58) | 230 (9.05) b | 363 (13.49) c | |

[#]p value from overall Chi-Square test when comparing food security status (food secure, at risk of hunger and experiencing hunger) across categorical variables (gender, race, marital status, educational attainment, source of income and locality), of which all were statistically significant.

[*]p<0.001 for age when comparing food secure vs at risk of hunger and food secure vs experiencing hunger (Kruskal-Wallis with Bonferroni correction). On multiple 2 by 2 analyses with Bonferroni correction, statistically significant differences within categories of sociodemographic characteristics are indicated by sharing the same letter: a, b, c, d or e, when comparing food secure vs at risk of hunger and food secure vs experiencing hunger. Specifically, p<0.01 between food secure vs. experiencing hunger except for:

[‖]Primary vs no schooling, salaries vs products/services; pensions/grants vs product/services and no income; p<0.01 between food secure vs. at risk of hunger except for:

[†]Coloured vs White/Indian/Asian; primary vs secondary; primary vs no schooling and secondary vs no schooling; salary vs product/services, pension/grants vs product/services, pension/grants vs no income and product /services vs no income; p<0.01 between food secure vs at risk of hunger and food secure vs experiencing hunger for married vs never married and married vs widower; p<0.01 between food secure vs experiencing hunger for married vs living together.

## Factors associated with experiencing hunger and being at risk of hunger in all households with and without children

Weighted univariate multinomial logistic regression (S1 Table) of all the households exhibits that household heads who were female, of older age, unmarried/widowed/separated vs married

were at higher risk of hunger and of experiencing hunger. However, when performing a weighted multivariable multinomial logistic regression (Fig 2A and 2B), age was no longer associated with experiencing hunger. Yet, female-headed households had 1.53 times increased odds of experiencing hunger as compared to male-headed households (p<0.001). In addition, having a larger household and a household with children present were also predictors of experiencing hunger; AOR = 1.06 (95% CI: 1.01–1.21); 1.68 (95% CI: 1.12–2.53) respectively. Having a household head who was either Coloured or White/Indian/Asian as opposed to African conferred lower odds of both experiencing hunger AOR = 0.29 (95% CI: 0.19–0.44) and 0.12 (95% CI: 0.04–0.33 and being at risk of hunger AOR = 0.54 (95% CI: 0.39–0.75), 0.22 (95% CI: 0.12–0.40. Furthermore, secondary/tertiary education conferred lower odds of both experiencing hunger; AOR = 0.40 (95% CI: 0.28–0.56 and being at risk of hunger; AOR = 0.69 (95% CI: 0.52–0.92. Households relying on pensions/social grants/remittances, or not having any source of income, were ~1.76 to 2.15 times more at risk of hunger and experiencing hunger than those receiving a salary/wage.

### Female household heads and food security in households with children

Compared to male heads of households with children (Table 3), female household heads were older, predominantly African, never married/widowed, have primary/secondary education or no schooling, reside in rural informal settings and have no income or rely on social grants/pensions/remittances. Lastly, male-headed households had more adults in the household while female-headed households had more children of all ages.

In the univariate analysis of the gender of the household head and food security in households with children, female-headed households had greater odds of being at risk of hunger [Odds Ratio (OR) = 1.58, (95% CI: 1.22–2.04), p<0.001] and experiencing hunger [OR = 2.35, (95% CI: 1.86–2.98), p<0.001].

### Child dependency and sociodemographic characteristics of household heads, in households with children

In the univariate analysis of child dependency and food security in households with children, having at least one child who was 15 years or older increased the odds of experiencing hunger by 28% (OR = 1.28 (95% CI: 1.06–1.56), p = 0.012). Table 4 shows associations between food security and child dependency, adjusted for the sociodemographic characteristics of the household heads. Compared to households with younger children only, having at least one child aged 15–19 years old in a household was not associated with either being at risk of hunger or experiencing hunger (no significant change in adjusted odds ratio). Having a household head who was male, Coloured/White/Indian/Asian, employed and having obtained tertiary education/higher degrees conferred lower odds of experiencing hunger. Residing in informal settings increased the odds of both being at risk of hunger and experiencing hunger.

## Discussion

To the best of our knowledge, the present study is the first to objectively quantify the burden of food insecurity among children and adolescents in South Africa and draw associations using a large-scale, population-based sample of households with children. No studies in South Africa have disaggregated households with and without children aged 0–19 years old thus possibly underestimating the true extent of childhood food insecurity and limiting direct comparisons to the present study. Indeed, there is an observed increase in food insecurity (experiencing hunger—25.9% to 32.5%) by 6.6% between the prevalence estimate reported for 2008 by Labadarios et al. [22] and the estimate generated in the present study. This lack of

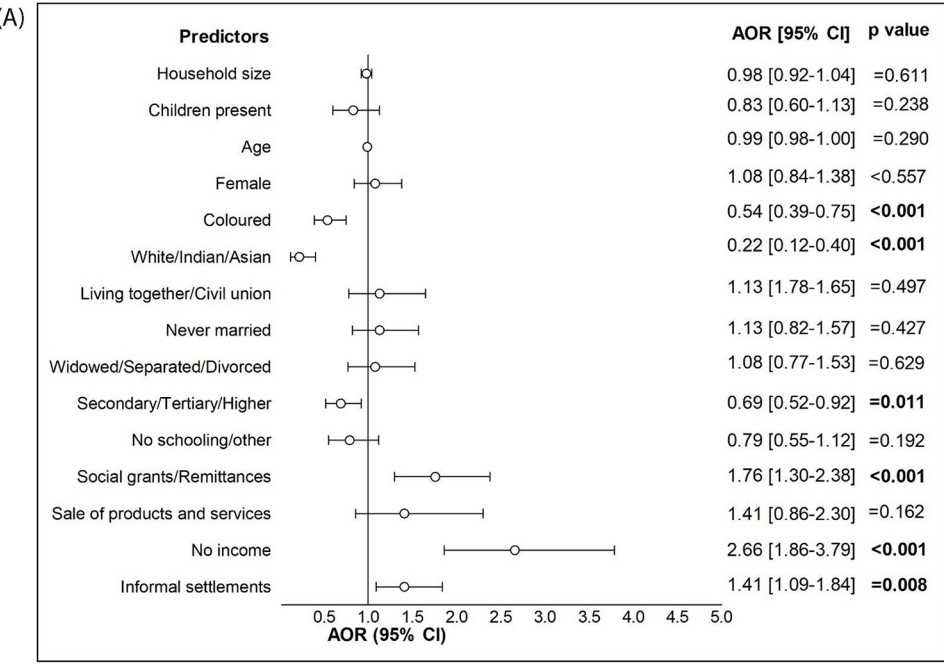

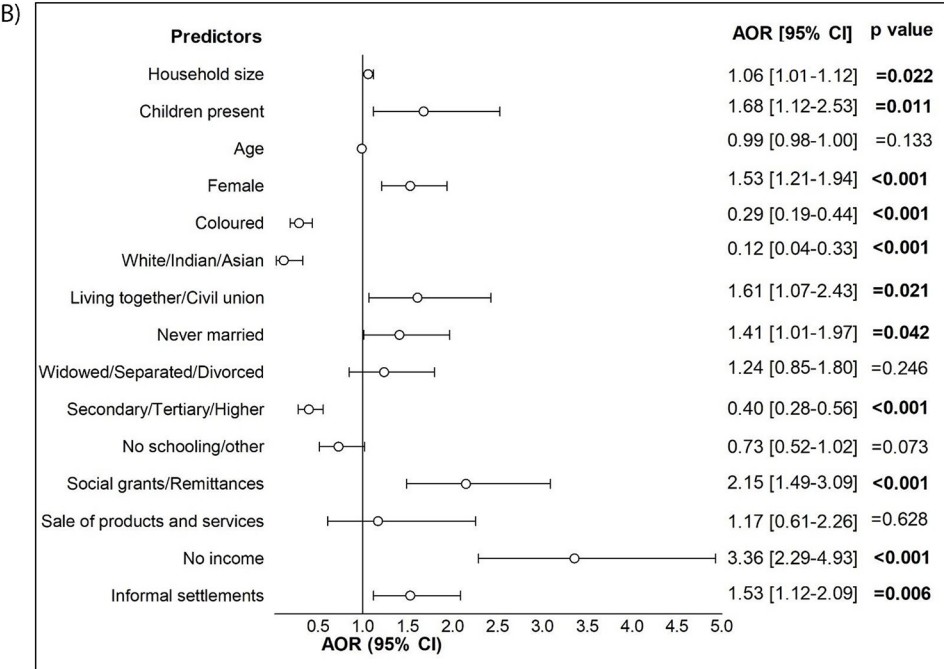

**Fig 2.** A. Factors associated with being at risk of hunger, data presented as adjusted odds ratios (AOR) and 95% Confidence intervals (95% CI). **Reference categories:** *Presence of children (no children present), Gender (male), Race (African), Marital status (married), Educational attainment (primary), Source of income (salaries/wages), Locality (formal).* Factors associated with experiencing hunger, data presented as adjusted odds ratios (AOR) and 95% Confidence intervals (95% CI). **Reference categories:** *Presence of children (no children present), Gender (male), Race (African), Marital status (married), Educational attainment (primary), Source of income (salaries/wages), Locality (formal).*

**Table 3. Sociodemographic characteristics of male and female-headed households with children.**

| | Total N = 3 495 | Male headed | Female headed | p value |
|---|---|---|---|---|
| | | n = 1 532 | n = 1 963 | |
| **Age, years (n = 3 495)** | | | | |
| Mean ± SD | 3 495 | 48.43±13.86 | 49.81±15.24 | 0.0423 |
| **Race (n = 3 493)[&]** | | | | |
| African | 2 462 | 976 (78.99) | 1 486 (88.40) | 0.0001 |
| Coloured | 658 | 324 (11.73) | 334 (7.67) | |
| White/Indian/Asian | 373 | 231 (9.29) | 142 (3.94) | |
| **Marital status (n = 3 420)[#]** | | | | |
| Married | 1 677 | 1 078 (71.39) | 599 (29.84) | <0.001 |
| Living together/civil unions | 309 | 222 (13.94) | 87 (4.07) | |
| Never married | 757 | 132 (9.09) | 625 (32.63) | |
| Widowed | 521 | 53 (3.58) | 468 (25.35) | |
| Separated/divorced | 156 | 28 (1.99) | 128 (8.11) | |
| **Educational attainment (n = 3 495)[^]** | | | | |
| Primary | 820 | 307 (18.28) | 516 (26.96) | <0.001 |
| Secondary | 1 521 | 706 (44.03) | 815 (40.08) | |
| Tertiary/Higher degrees | 353 | 215 (16.24) | 138 (6.96) | |
| No schooling/other | 801 | 388 (21.45) | 494 (26.00) | |
| **Source of income (n = 3 116)[$]** | | | | |
| Salaries or wages | 1 328 | 798 (55.44) | 530 (29.45) | <0.001 |
| Social grants/pensions/Remittances | 1 006 | 305 (23.66) | 701 (43.58) | |
| Sale of products and services | 138 | 76 (5.46) | 62 (4.00) | |
| No income | 644 | 225 (15.45) | 419 (22.97) | |
| **Locality (n = 3 495)[*]** | | | | |
| Urban formal | 1 836 | 874 (60.13) | 962 (48.31) | <0.001 |
| Urban informal | 464 | 178 (8.28) | 286 (10.82) | |
| Rural formal | 359 | 191 (7.27) | 168 (4.44) | |
| Rural informal | 836 | 289 (24.32) | 547 (36.43) | |
| **Composition, Median [IQR]** | | | | |
| Household size | 3 495 | 5 [4–6] | 5 [3–6] | 0.090 |
| No. of adults | 3 495 | 3 [2–4] | 2 [2–3] | <0.001 |
| No. of children | 3 495 | 2 [1–3] | 2 [1–4] | <0.001 |
| No. of children aged 0–14 years | 1814 | 2 [1–3] | 2 [1–3] | <0.001 |
| No. of children aged 15–19 years | 1681 | 2 [2–4] | 3 [2–4] | <0.001 |

[&]Race: African vs Coloured (p = 0.001) and African vs White/Indian/Asian (p = 0.001)

[#]Marital status: married vs never married (p<0.001), married vs widowed (p<0.001), married vs separated/divorced (p<0.001), living together/civil union vs never married (p<0.001), living together/civil union vs widowed (p<0.001), living together/civil union vs separated/divorced (p<0.001), and never married vs widowed (p = 0.0015)

[^]Educational attainment: primary vs secondary (p = 0.0002), primary vs tertiary/higher degrees (p<0.001), secondary vs tertiary/higher degrees (p = 0.0001), and tertiary vs no schooling (p<0.001)

[$]Source of income: salaries/wages vs social grants/pensions/remittances (p<0.001), salaries/wages vs no income (p<0.001), and social grants/pensions/remittances vs sale of products and services (p = 0.0002)

[*]Locality: urban formal vs urban informal (p = 0.001), urban formal vs rural informal (p<0.001), urban informal vs rural informal (p = 0.0010), and rural formal vs rural informal (p<0.001).

**Table 4. Weighted multivariable multinomial regression analysis between food security and child dependency, adjusted for sociodemographic characteristics of the heads of households with children.**

| Characteristics | At risk of hunger | | p value | Experiencing hunger | | |
|---|---|---|---|---|---|---|
| | AOR | 95% CI | | AOR | 95% CI | p value |
| Household size | 0.98 | 0.92–1.04 | 0.604 | 1.04 | 0.99–1.10 | 0.102 |
| **Child dependency** | | | | | | |
| Children aged 0–14 years only | *REF* | *REF* | *REF* | *REF* | *REF* | *REF* |
| At least one child aged 15–19 years | 1.01 | 0.78–1.30 | 0.934 | 1.24 | 0.95–1.62 | 0.105 |
| **Age, years** | | | | | | |
| Age | 0.99 | 0.98–1.01 | 0.336 | 0.98 | 0.97–1.00 | 0.089 |
| **Gender** | | | | | | |
| Female | *REF* | *REF* | *REF* | *REF* | *REF* | *REF* |
| Male | 0.92 | 0.66–1.27 | 0.603 | 0.72 | 0.53–0.98 | 0.041 |
| **Race** | | | | | | |
| African | *REF* | *REF* | *REF* | *REF* | *REF* | *REF* |
| Coloured | 0.60 | 0.42–0.87 | 0.006 | 0.33 | 0.21–0.50 | <0.001 |
| White/Indian/Asian | 0.17 | 0.09–0.34 | <0.001 | 0.12 | 0.04–0.34 | <0.001 |
| **Marital status** | | | | | | |
| Living together/civil union | *REF* | *REF* | *REF* | *REF* | *REF* | *REF* |
| Married | 0.88 | 0.59–1.32 | 0.552 | 0.61 | 0.39–0.94 | 0.027 |
| Never married | 0.88 | 0.51–1.15 | 0.652 | 0.83 | 0.51–1.36 | 0.478 |
| Widowed | 0.67 | 0.37–1.19 | 0.172 | 0.63 | 0.35–1.12 | 0.117 |
| Separated/divorced | 1.28 | 0.59–2.75 | 0.526 | 1.83 | 0.54–2.56 | 0.673 |
| **Educational attainment** | | | | | | |
| No schooling/other | *REF* | *REF* | *REF* | *REF* | *REF* | *REF* |
| Primary | 1.23 | 0.82–1.83 | 0.310 | 1.37 | 0.94–1.99 | 0.095 |
| Secondary | 1.06 | 0.72–1.58 | 0.742 | 0.70 | 0.46–1.05 | 0.086 |
| Tertiary/Higher degree | 0.50 | 0.28–0.89 | 0.019 | 0.18 | 0.09–0.34 | <0.001 |
| **Source of income** | | | | | | |
| No income | *REF* | *REF* | *REF* | *REF* | *REF* | *REF* |
| Pensions/Grants/Remittances | 0.91 | 0.60–1.36 | 0.651 | 0.82 | 0.55–1.21 | 0.331 |
| Sale of products and services | 0.73 | 0.40–1.34 | 0.323 | 0.40 | 0.19–0.81 | 0.011 |
| Salaries and/or wages | 0.51 | 0.34–0.77 | 0.001 | 0.38 | 0.25–0.57 | <0.001 |
| **Locality** | | | | | | |
| Formal | *REF* | *REF* | *REF* | *REF* | *REF* | *REF* |
| Informal | 1.57 | 1.15–2.15 | 0.004 | 1.66 | 1.18–2.34 | 0.003 |

improvement may be accounted for by the enduring socio-political and economic challenges sustained since the end of apartheid [22, 38]. Drawing from a cross-sectional, population-based national sample of South African households, we have shown that the prevalence of food insecurity in households with children was nearly 60%. This is similar to the original SAN-HANES study where food insecurity among all households was 54% [12]. In the present study, after adjusting for household size, the sociodemographic characteristics of the household head and the main source of income, the presence of children and adolescents conferred ~1.68 increased odds of food insecurity (experiencing hunger) relative to households without children. In addition, having a female vs male household head, having an African household head compared to all other race groups, living in informal settings vs formal settings and having no income compared to relying on salaries or wages increased the odds of experiencing hunger between 53% and 300%. Secondary/tertiary/higher educational attainment of the household

head reduced the odds of experiencing hunger by at least half relative to primary education. Reliance on social grants, pensions and remittances as the main source of income was associated with more than double the odds of being food insecure. When analysing food security in households with at least one adolescent aged 15–19 years, versus those with children aged 0–14 years only, before adjusting for the sociodemographics of the household head and location, the odds of being food insecure were almost 30% higher compared to households with younger children only. However, after controlling for these factors, neither child dependency nor the ability to work were associated with any category of food security.

## Household head sociodemographics and household characteristics

The majority of households with children and adolescents in our sample was headed by women, had African household heads, or were living in urban or rural informal areas. Therefore, the increased vulnerability of households with children may be explained, in part, by the additive effects of these known predictors of food insecurity [39]. Previous studies in other contexts have found that the presence of children in households is independently associated with food insecurity, beyond sociodemographic characteristics and measures of socioeconomic status [7, 39–43]. Additional research has demonstrated that the sex of the household head is an important factor predicting household vulnerabilities, with the highest prevalence of hunger reported among female-headed households [27], a risk that worsens with rural location [25, 44]. Our comparative analysis of female- versus male-headed households points to the stark sociodemographic disadvantages that women heads endure in households with children, confirming previous research in South Africa on the gendered nature of poverty and food insecurity [27, 45]. In their large-scale, population-based longitudinal analysis, the Indonesian Family Life Survey, Vaezghasemi et al. [46] further argue that although women may be conscious of healthy food options, because they are at the bottom of the family hierarchy and have low social capital, they remain defenseless against household hunger. Given South Africa's colonial and apartheid history, racial differences still persist in the prevalence of food insecurity, with black or African households bearing the greatest burden compared to historically more advantaged minority groups in the country [7, 11, 12, 25].

Previous studies have demonstrated strong positive associations between the lowest levels of educational attainment and food insecurity, which can be mitigated by higher levels of educational attainment [5, 47–50]. Likewise, we found that improved levels of education (secondary and tertiary educational attainment) reduced the risk of food insecurity by at least half; yet, having no formal schooling was not associated with food insecurity. As Chakona and Shackleton also found [25], we suggest that household heads without a formal education may rely on subsistence farming or implement other skills-based practices which might act as a safety net against hunger.

The Born in Bradford cohort, a longitudinal study in the United Kingdom, showed associations between cohabitation status and food insecurity, where pregnant women not living with a partner had a two-fold increased risk of food insecurity [51]. Hanson, Sobal, and Frongillo [52] further demonstrated that marital status was associated with food insecurity among men, but not women. We found significant associations between the marital status of the household head and risk of experiencing hunger. Even after adjustments for sex, living together/civil unions and never married increased the odds of experiencing hunger by 61% and 41%, respectively.

## Main source of household income, social grants and child dependency

Using self-reported main source of income to indicate a household's economic stability, we note that there were no differences in income sources between households with and without

children. However, household size was larger in households with children, including an increased number of adults. We show that not having any source of income predicts that a household will experience a three-fold increase in experiencing hunger. Furthermore, similar to Ruiters and Wildschutt [27], receiving government social grants, pensions or remittances did not alleviate hunger nor the risk of being food insecure for households with and without children. Despite a substantial social safety net wherein 16.2 million South Africans in 2013, representing nearly one third of the country's population received different types of government grants [53], these were not protective in households with children, with a 76% increase in being at risk of food insecurity and double the odds for being food insecure. Other contexts show mixed results for government assistance programmes on alleviating food insecurity. Tarasuk, Fafard St-Germain and Mitchell. [54] have shown that in Canada, household before-tax income adjusted for family size was protective against food insecurity; however, households receiving social assistance were three times more likely to be food insecure. In contrast, Brown and Tarasuk. [55] showed a significant decline in severe food insecurity subsequent to the roll-out of the Canada Child Benefit (CCB) in households with children, a country-wide non-means-tested cash transfer programme. In Brazil, participation in the Bolsa Familia Programme [56] and in the United States receiving Supplemental Nutrition Assistance Program (SNAP, formerly the Food Stamp Program) support [8, 57], were not effective in preventing food insecurity. Furthermore, dietary quality for children and adolescents receiving SNAP benefits did not improve [58]. In one South African study where social grants did show improvements in household food insecurity and dietary quality, the authors note that these findings were not accompanied by concomitant improvements in anthropometric indicators of child malnutrition [26]. Even when controlling for the sociodemographic characteristics of the household head and the main source of income, the risk of experiencing hunger was increased with the presence of children in all households. This is similar to the findings of a 2017 national study comparing households with and without children under the age of 5 years, whereby households with no children were less likely to experience inadequate access to food [7]. In our sample of households with children, controlling for child dependency status (ages 0–14 versus 15–19 years old) did not lessen the odds of experiencing hunger. Regarding an older child's potential to contribute towards household income through work eligibility, Stats-SA reported in 2012, the year of the SANHANES-1, an overall youth unemployment rate of 35.8% (age 15–34 years old) as compared to an adult unemployment rate of 15.1% [53]. After disaggregating youth unemployment by age and sex, males and females aged 15–19 years had rates of 50.2% versus 74% respectively [53], indicating that child dependency on an adult household head extends throughout adolescence in South Africa. Such continued dependence on an adult with resultant food insecurity has repercussions for the psychosocial functioning of teenagers, as it predicts future behavioural problems, directly and indirectly, through parental caregiver mental health stressors [59].

## Contemporary considerations and current context

The present COVID-19 pandemic, occurring alongside household and child food and nutrition insecurity, presents a disconcerting syndemic [60] that is capable of reversing any recent mitigation of hunger at household and individual levels in the country [60]. Indeed, the COVID-19 pandemic has forced South Africa into an abrupt and prolonged lockdown threatening job and societal security [61]. Data from the National Income Dynamics Study-Coronavirus Rapid Mobile Survey (NID-CRAMS) [62], a four-wave longitudinal survey of adult South Africans, indicate that even with re-opening of the economy, lifting restrictions on movement and returning to work, 37–47% households did not have sufficient money to buy

food, with 16–22% stating that they experienced hunger, during the period from May/June 2020 to February/March 2021 [62]. Overall, while hunger declined somewhat over this period, the poorest quintile suffered more hunger than before [62]. Without augmented social supports for households with children and adolescents, and an emphasis on nutrition-sensitive interventions [29, 50, 63], it is likely that food insecurity will pervade [14] and possibly impede the country from realising its 2030 Sustainable Development Goal of achieving zero hunger [31, 60].

## Limitations

The present study is not without limitations. The SANHANES-1 was conducted in 2011/12. Although never repeated, the data are at least nine years old and may not reflect the current status of food security in South Africa. Nonetheless, the SANHANES provides the most recent and comprehensive dataset on food security at the national and household-level using the CCHIP index that explores the context and loci within which food insecurity exists. This is in contrast with other national surveys which only record the frequency of problems satisfying food needs "in the past 12 months" [13]. Secondly, a comprehensive assessment of household socioeconomic status was beyond the scope of this research. Having considered the primary source of household income and educational status as contributing factors to household economic stability, a more robust measure of socioeconomic status using household assets could add a more granular dimension to the analysis.

## Conclusion

Our study confirms that South African children and adolescents face a major challenge of food insecurity that is driven by a number of household head characteristics and household location. The untoward effects of food insecurity on child and adolescent health are not evanescent and have far-reaching adverse health outcomes. Therefore, this is a clarion call to bolster multi-sectoral interventions and healthy public policies, with a strong focus on the social determinants of health and poverty eradication strategies targeting these vulnerable households.

## Supporting information

**S1 Fig. Database management, organisation and integration of data extracted from the four household and individual—level datasets of the SANHANES-1.**
(DOCX)

**S1 Table. Weighted univariable multinomial logistic regression analysis of factors associated with being at risk of hunger and experiencing hunger against being food secure in households with and without children.**
(DOCX)

**S1 Dataset. Final analytic sample of households with food security categories.**
(CSV)

## Acknowledgments

The authors acknowledge Dr Innocent Maposa for his assistance with data analysis.

## Author Contributions

**Conceptualization:** Siluleko Mkhize, Elena Libhaber, Laurel Baldwin-Ragaven.

**Data curation:** Siluleko Mkhize, Ronel Sewpaul, Priscilla Reddy.

**Formal analysis:** Siluleko Mkhize, Elena Libhaber, Ronel Sewpaul, Laurel Baldwin-Ragaven.

**Investigation:** Siluleko Mkhize, Elena Libhaber, Ronel Sewpaul, Priscilla Reddy, Laurel Baldwin-Ragaven.

**Methodology:** Siluleko Mkhize, Elena Libhaber, Laurel Baldwin-Ragaven.

**Project administration:** Ronel Sewpaul, Priscilla Reddy.

**Supervision:** Elena Libhaber, Laurel Baldwin-Ragaven.

**Visualization:** Siluleko Mkhize, Elena Libhaber.

**Writing – original draft:** Siluleko Mkhize, Elena Libhaber, Laurel Baldwin-Ragaven.

**Writing – review & editing:** Siluleko Mkhize, Elena Libhaber, Ronel Sewpaul, Priscilla Reddy, Laurel Baldwin-Ragaven.

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
