## [Decision Letter · Decision Letter 0]

26 Apr 2021

PONE-D-20-39991

Childhood food insecurity in South Africa: A household-level analysis of hunger

PLOS ONE

Dear Dr. Mkhize,

Thank you for submitting your manuscript to PLOS ONE. After careful consideration, we feel that it has merit but does not fully meet PLOS ONE’s publication criteria as it currently stands. Therefore, we invite you to submit a revised version of the manuscript that addresses the points raised during the review process.

We look forward to receiving your revised manuscript.

Kind regards,

Rubeena Zakar, Ph.D

Academic Editor

PLOS ONE

Journal Requirements:

[SM acknowledges the South African National Research Foundation for financial support of his
postgraduate studies which enabled him to pursue this research.]

 [The author(s) received no specific funding for this work]

Reviewers' comments:

Reviewer's Responses to Questions

**Comments to the Author**

1. Is the manuscript technically sound, and do the data support the conclusions?

Reviewer #1: Yes

Reviewer #2: No

2. Has the statistical analysis been performed appropriately and rigorously? 

Reviewer #1: Yes

Reviewer #2: Yes

3. Have the authors made all data underlying the findings in their manuscript fully available?

Reviewer #1: Yes

Reviewer #2: No

4. Is the manuscript presented in an intelligible fashion and written in standard English?

Reviewer #1: Yes

Reviewer #2: Yes

5. Review Comments to the Author

Reviewer #1: Since the previous research studies related to food insecurity lack a specific focus on households with children, so I found the study worth reading. A comparative analysis of different household characteristics between households with and without children is made. Various descriptive and inferential statistical methods have been used by the authors to analyze the data. The findings of the study are quite interesting and believable. The language and draft of the paper is also excellent.

Reviewer #2: Generally, I found this manuscript to be well written. However, the relevance to current circumstances is questionable and the findings are not adequately explored.

I provide specific comments below, if you choose to make revisions.

The SANHANES-1 dataset has been used for many publications in the academic literature. The introduction needs to better summarise previous research and identify the gap in the literature that this study seeks to fill. Please reference Shisana and colleagues 2013 report at the very least. An early blog post by Parker in 2013 on the data may also be relevant.

Lines 92 – 20 years of age seems like an arbitrary cut-off for the definition of a child.

Lines 96 & 419 refers to improvements between 2002 and 2017. Source and further information is needed.

Line 138 introduces the concept of nutrition-sensitive interventions without any definition, background or references. Consider adding a paragraph summarising nutrition-sensitive interventions generally and in South Africa and how sociodemographic correlates can inform the design of these interventions to improve childhood health.

Lines 140-140 The sentence spanning these lines is repeating information from earlier in the introduction. Rephrase.

Lines 198-199 The sentence spanning these lines is superfluous. This is rudimentary data manipulation and is expressed in the previous sentence.

Lines 273-275 This is unclear. What were the differences?

Lines 287, 290, 305, 312 Identifying factors as being directly ‘protective for experiencing hunger’ is misleading. Many of these factors are likely causally related to asset endowment and/or income. They are not protective of but negatively related to. However, this term is appropriate when discussing social grants – that are designed to be protective against extreme poverty and food insecurity.

Lines 312-313 The sentence spanning these lines seems more emotive than the rest of the results section.

A theme throughout the manuscript is this purported influence of having children on food insecurity. This appears twice in the introduction, is expressed as a poignant point in the results section and in discussion it is suggested that there is a mounting body of evidence on the topic. The evidence as presented in this manuscript is weak in supporting the assertion “children below age 18 years in households is independently associated with food insecurity, beyond sociodemographic characteristics and measures of socioeconomic status”.

The most interesting finding in this manuscript is that “social grants and pensions, or remittances, did not alleviate hunger or the risk of being food insecure.” This needs to be analysed and discussed further.

The link between this manuscript and the COVID-19 pandemic is tenuous. The authors have done very little to link the dated dataset to present day.

Inconsistency in p value presentation.

Original sources need to be cited.

6. PLOS authors have the option to publish the peer review history of their article (what does this mean?). If published, this will include your full peer review and any attached files.

Reviewer #1: **Yes: **Muhammad Azeem

Reviewer #2: No

---

## [Author Response · Author response to Decision Letter 0]

25 Jun 2021

See attached "Response to Reviewers" file

---

## [Decision Letter · Decision Letter 1]

23 Jul 2021

PONE-D-20-39991R1

Childhood food insecurity in South Africa: A household-level analysis of hunger

PLOS ONE

Dear Dr. Mkhize,

Thank you for submitting your manuscript to PLOS ONE. After careful consideration, we feel that it has merit but does not fully meet PLOS ONE’s publication criteria as it currently stands. Therefore, we invite you to submit a revised version of the manuscript that addresses the points raised during the review process.

We look forward to receiving your revised manuscript.

Kind regards,

Rubeena Zakar, Ph.D

Academic Editor

PLOS ONE

Reviewers' comments:

Reviewer's Responses to Questions

**Comments to the Author**

1. If the authors have adequately addressed your comments raised in a previous round of review and you feel that this manuscript is now acceptable for publication, you may indicate that here to bypass the “Comments to the Author” section, enter your conflict of interest statement in the “Confidential to Editor” section, and submit your "Accept" recommendation.

Reviewer #2: (No Response)

Reviewer #3: (No Response)

2. Is the manuscript technically sound, and do the data support the conclusions?

Reviewer #2: No

Reviewer #3: Yes

3. Has the statistical analysis been performed appropriately and rigorously? 

Reviewer #2: Yes

Reviewer #3: Yes

4. Have the authors made all data underlying the findings in their manuscript fully available?

Reviewer #2: Yes

Reviewer #3: Yes

5. Is the manuscript presented in an intelligible fashion and written in standard English?

Reviewer #2: Yes

Reviewer #3: Yes

6. Review Comments to the Author

Reviewer #2: The revisions to the manuscript show improvement. Background detail on nutrition sensitive interventions has been provided adequately, the link with this study and the COVID-19 pandemic is better supported and some revisions have been made to improve the scientific writing. However, there are still substantial deficiencies in the interpretation of the literature and the interpretation of results. In general I find that the language used in the manuscript suggests that causality has not been appropriately interpreted. In addition, my initial points still stand on 1) the purported influence of having children on food insecurity and 2) social grants and pensions, or remittances.

The classification of children being under 0-19 is problematic. In the introduction, the authors emphasise the implications of household food insecurity among children, where the implications of undernutrition for wasting and stunting are most relevant for young children, less so for adolescence. Given the age threshold, this background information is irrelevant to the present study. Further, the cited studies do not justify the threshold used in this study. The present study does not assess BMI or other anthropometric measurements. Rather than appropriating this threshold for anthropometry it is more relevant to set a threshold based on child dependency, where in this analysis, some ‘children’ will be an important source of labour for the household and may even support other “mouths to feed”. Most critically, the talking point remains in the discussion about the “pressure of mouths to feed”. The authors acknowledge that “the majority of households with children in our sample were headed by women… vulnerability of households with children may be explained, in part, by the additive effects”. Yet, the authors still attempt to add potential Type I errors to a supposed “mounting body of evidence”. Revisit the analysis in order to 1) better account for the gender of household head and child association and 2) assess the sensitivity of your results to age - using thresholds more relevant to dependency on a carer.

The authors need to be cognisant of the causality of the association between social grants (etc) and food insecurity. These data were not collected as part of a randomised control trial designed to assess the effectiveness of social grants. A mistaken view of causality pervades the language of the manuscript. Furthermore, the introduction of the manuscript contains the following: “Households that are reliant on private food charities and government assisted social grants are at an three-fold increased risk of food insecurity”. This is based on mistaken causality and a misuse of a descriptive study (Ruiters and Wildschutt 2010). The study that the authors cite, in my opinion, contained ill-informed suggestions of reallocating social grants from food insecurity prevention to lump-sum expenditures and that “Food baskets could be provided to families in exchange for work” – that is demeaning. This is only one example of an inadequate interpretation of the literature.

Reviewer #3: The manuscript looks okay but with minor revision.

L141-143: Any reason found in the literature for this. It will be nice if you add few reasons as being found in the literature. If there is none, it may be stated.

L202-204:Was the ethical clearance obtained for the use of the secondary data or this was the ethical clearance obtained for the collection of the primary data? If the latter holds, please re-phrase to indicate that.

L207: Household-level or Household level? Please check the entire manuscript and be consistent with one.

L227: Is it arithmetic or geometric mean? Please indicate?

Please check for other comments that were inserted directly in the manuscript.

7. PLOS authors have the option to publish the peer review history of their article (what does this mean?). If published, this will include your full peer review and any attached files.

Reviewer #2: No

Reviewer #3: **Yes: **Alamu Emmanuel Oladeji(PhD, FIFST, MNIFST)

---

## [Author Response · Author response to Decision Letter 1]

16 Jan 2022

The Response to Reviewers document has been uploaded and labelled as such.

---

## [Decision Letter · Decision Letter 2]

28 Mar 2022

PONE-D-20-39991R2Childhood food insecurity in South Africa: A household-level analysis of hungerPLOS ONE

Dear Dr. Baldwin-Ragaven

Thank you for submitting your manuscript to PLOS ONE. After careful consideration, we feel that it has merit but does not fully meet PLOS ONE’s publication criteria as it currently stands. Therefore, we invite you to submit a revised version of the manuscript that addresses the points raised during the review process.

We look forward to receiving your revised manuscript.

Kind regards,

Rubeena Zakar, Ph.D

Academic Editor

PLOS ONE

Reviewers' comments:

Reviewer's Responses to Questions

**Comments to the Author**

1. If the authors have adequately addressed your comments raised in a previous round of review and you feel that this manuscript is now acceptable for publication, you may indicate that here to bypass the “Comments to the Author” section, enter your conflict of interest statement in the “Confidential to Editor” section, and submit your "Accept" recommendation.

Reviewer #2: (No Response)

Reviewer #3: All comments have been addressed

2. Is the manuscript technically sound, and do the data support the conclusions?

Reviewer #2: Partly

Reviewer #3: Yes

3. Has the statistical analysis been performed appropriately and rigorously? 

Reviewer #2: Yes

Reviewer #3: Yes

4. Have the authors made all data underlying the findings in their manuscript fully available?

Reviewer #2: Yes

Reviewer #3: Yes

5. Is the manuscript presented in an intelligible fashion and written in standard English?

Reviewer #2: Yes

Reviewer #3: Yes

6. Review Comments to the Author

Reviewer #2: Some of these revisions have, indeed, improved this manuscript - particularly in incorporating Table 3 and Table 4. Unfortunately, my underlying concerns remain.

Reviewer #3: Thanks to the authors for responding to all comments satisfactorily. However, please check L-187, a reference is missing, and please add the reference.

7. PLOS authors have the option to publish the peer review history of their article (what does this mean?). If published, this will include your full peer review and any attached files.

Reviewer #2: No

Reviewer #3: **Yes: **Alamu, Emmanuel Oladeji(Ph.D., FIFST, MNIFST)

---

## [Author Response · Author response to Decision Letter 2]

16 May 2022

Kindly see attached "Response to Reviewers" file.

---

## [Decision Letter · Decision Letter 3]

1 Sep 2022

PONE-D-20-39991R3Childhood food insecurity in South Africa: A household-level analysis of hungerPLOS ONE

Dear Dr. Baldwin-Ragaven,

Thank you for submitting your manuscript to PLOS ONE. After careful consideration, we feel that it has merit but does not fully meet PLOS ONE’s publication criteria as it currently stands. Therefore, we invite you to submit a revised version of the manuscript that addresses the points raised during the review process.

We look forward to receiving your revised manuscript.

Kind regards,

Rubeena Zakar, Ph.D

Section Editor

PLOS ONE

Journal Requirements:

Reviewers' comments:

Reviewer's Responses to Questions

**Comments to the Author**

1. If the authors have adequately addressed your comments raised in a previous round of review and you feel that this manuscript is now acceptable for publication, you may indicate that here to bypass the “Comments to the Author” section, enter your conflict of interest statement in the “Confidential to Editor” section, and submit your "Accept" recommendation.

Reviewer #3: All comments have been addressed

Reviewer #4: (No Response)

Reviewer #5: (No Response)

Reviewer #6: (No Response)

2. Is the manuscript technically sound, and do the data support the conclusions?

Reviewer #3: Yes

Reviewer #4: Yes

Reviewer #5: Yes

Reviewer #6: (No Response)

3. Has the statistical analysis been performed appropriately and rigorously? 

Reviewer #3: Yes

Reviewer #4: Yes

Reviewer #5: No

Reviewer #6: (No Response)

4. Have the authors made all data underlying the findings in their manuscript fully available?

Reviewer #3: Yes

Reviewer #4: Yes

Reviewer #5: Yes

Reviewer #6: (No Response)

5. Is the manuscript presented in an intelligible fashion and written in standard English?

Reviewer #3: Yes

Reviewer #4: Yes

Reviewer #5: No

Reviewer #6: Yes

6. Review Comments to the Author

Reviewer #3: Excellent effort to respond to all the comments has improved the manuscript tremendously. Please see the minor edits and comments inserted in the manuscript for your consideration.

Reviewer #4: Summary

Thank you for the opportunity I have to provide a review for this important article on Childhood food insecurity in South Africa. The authors have done a good job of identifying the extent of childhood hunger and factors which contribute to it.

Title

• The title clearly depicts the research that was carried out with focus on the outcome, however the period or time of investigation is missing

• Each author’s contribution was clearly stated, and corresponding author’s contact is clear.

Abstract

• The abstract is unstructured, and it allowed for a good read. It is a good summary of the workdone.

Introduction

• The introduction revealed the intent of the study and raise curiosity as to what could be the finding of the study.

• However, the authors used very old references-I think they could use more recent references. I could not relate well with when the study weas conducted because I could not find anywhere it was mentioned. I could have missed it.

• The objectives were well stated: The authors wanted to determine the prevalence of food insecurity in households with and without children as well as to assess associations between the sociodemographic characteristics of the household head and degrees of hunger.

Methods and Materials

• The methods were well explained, and it is repeatable.

• The SANHANES-1 database was used for this study, however the question is when was the data extracted or when was this study conducted?

• Ethical consideration should have been the last in method. Authors can consider making the changes if they want.

Results

• Well written results and clearly stated

• In lines 354-6 – larger household as a predictor of experiencing hunger seems not to be related. LCL of 1.00 shows that if repeated a researcher can get AOR of 1.00. I think 4 digits LCL should be used if not it should not be reported because it is an unlikely association.

Discussion

• The discussion is well written and provides the necessary reasoning behind the results of the study

• It is was nice that they provided sub-headings which made an easy understanding of the discussion.

• The limitation of the study was clearly stated which eventually clarifies some of my earlier questions on when this study was conducted. But I think the authors should clearly state the year of study in the title.

Conclusion

• Provides answers to the objectives of the study

• The recommendations made by the authors may not be relevant since the authors agreed that this is a very old data, but the only data available for this kind of secondary data analysis.

Recommendation

• Accept with minor revision

Reviewer #5: It is an article with clear objective and met it if the method of analysis used was correct.

But I doubt on the selection of method for analysis. Is it difficult to dichotomize the outcome variable? If so, why you did not use another method of data analysis? Because, multinomial logistic regression is usually not preferred except when the the outcome variable is difficult to dichotomize or the categories of the outcome variable is worth as it is or multinomial.

Reviewer #6: Review of the Paper “Childhood food insecurity in South Africa: A household-level analysis of hunger”

The title is better to be “prevalence of food in security and associated factors among households in South Africa” Or “Childhood and adolescent food insecurity in south Africa”

The gap to be filled by this study should be clearly stated, such as the variables added other than other similar studies, method change or modification etc.

The AOR better to precede the CI like AOR=1.68[1.12-2.53] with 95%CI.

Instead of saying lower odds of experiencing hunger, I recommend to say there was a preventive/protective risk for hunger for those AOR<1

Please minimize the key words

The table titles don’t contain time period

In the discussion part the implication of the study should be stated after comparing and contrasting.

The conclusion is better to declare that food insecurity (experiencing hunger), at risk of hunger and food secured were low, medium or high.

Regardless of these minor corrections the paper is very interesting and informative. It is well organized; better to be published.

Thank you!!!

7. PLOS authors have the option to publish the peer review history of their article (what does this mean?). If published, this will include your full peer review and any attached files.

Reviewer #3: **Yes: **Alamu Emmanuel Oladeji (PhD, IFST, MNIFST)

Reviewer #4: **Yes: **ADEBAYO PETER ADEWUYI

Reviewer #5: No

Reviewer #6: **Yes: **Wondimnew Desalegn Addis Lecturer of Epidemiology at Debre Tabor University, Ethiopia.

---

## [Author Response · Author response to Decision Letter 3]

1 Oct 2022

Please see attached "Response to Reviewers" PDF document where we have addressed all reviewers' concerns. Thank you.

---

## [Decision Letter · Decision Letter 4]

14 Nov 2022

Child and adolescent food insecurity in South Africa: A household-level analysis of hunger

PONE-D-20-39991R4

Dear Dr. Baldwin-Ragaven, 

We’re pleased to inform you that your manuscript has been judged scientifically suitable for publication and will be formally accepted for publication once it meets all outstanding technical requirements.

Kind regards,

Rubeena Zakar, Ph.D

Section Editor

PLOS ONE

Additional Editor Comments (optional):

Reviewers' comments:

Reviewer's Responses to Questions

**Comments to the Author**

1. If the authors have adequately addressed your comments raised in a previous round of review and you feel that this manuscript is now acceptable for publication, you may indicate that here to bypass the “Comments to the Author” section, enter your conflict of interest statement in the “Confidential to Editor” section, and submit your "Accept" recommendation.

Reviewer #3: All comments have been addressed

Reviewer #6: All comments have been addressed

2. Is the manuscript technically sound, and do the data support the conclusions?

Reviewer #3: Yes

Reviewer #6: Yes

3. Has the statistical analysis been performed appropriately and rigorously? 

Reviewer #3: Yes

Reviewer #6: Yes

4. Have the authors made all data underlying the findings in their manuscript fully available?

Reviewer #3: Yes

Reviewer #6: Yes

5. Is the manuscript presented in an intelligible fashion and written in standard English?

Reviewer #3: Yes

Reviewer #6: Yes

6. Review Comments to the Author

Reviewer #3: The authors have done an excellent job by responding carefully to all the comments and this has improved the quality of the paper. However, I have minor edits for consideration by the authors.

Reviewer #6: Unless some minor errors like ediorial issues, it is nice work and Well done! for me now it is Ok to be published. Go ahead please. Thank you so much!

7. PLOS authors have the option to publish the peer review history of their article (what does this mean?). If published, this will include your full peer review and any attached files.

Reviewer #3: **Yes: **Alamu Emmanuel Oladeji

Reviewer #6: **Yes: **Wondimnew Desalegn Addis , Debre Tabor University, College of Health Sciences, Department of Public Health/Epidemiology,

---

## [Editor Report · Acceptance letter]

5 Dec 2022

PONE-D-20-39991R4 

Child and adolescent food insecurity in South Africa: A household-level analysis of hunger 

Dear Dr. Baldwin-Ragaven:

I'm pleased to inform you that your manuscript has been deemed suitable for publication in PLOS ONE. Congratulations! Your manuscript is now with our production department. 

Kind regards, 

on behalf of

Dr. Rubeena Zakar 

Section Editor

PLOS ONE